# Associations between Suboptimal Sleep and Smoking, Poor Nutrition, Harmful Alcohol Consumption and Inadequate Physical Activity (‘SNAP Risks’): A Comparison of People with and without a Mental Health Condition in an Australian Community Survey

**DOI:** 10.3390/ijerph18115946

**Published:** 2021-06-01

**Authors:** Alexandra P. Metse, Tara Clinton-McHarg, Elise Skinner, Yogayashwanthi Yogaraj, Kim Colyvas, Jenny Bowman

**Affiliations:** 1School of Psychology, University of Newcastle, University Drive, Callaghan, NSW 2308, Australia; tara.clinton-mcharg@newcastle.edu.au (T.C.-M.); elise.skinner@uon.edu.au (E.S.); jenny.bowman@newcastle.edu.au (J.B.); 2School of Health and Behavioural Sciences, University of the Sunshine Coast, 90 Sippy Downs Dr., Sippy Downs, QLD 4556, Australia; 3Hunter Medical Research Institute, Lot 1 Kookaburra Circuit, New Lambton Heights, NSW 2305, Australia; 4Discipline of Psychology, Murdoch University, 90 South Street, Murdoch, WA 6150, Australia; yashi.yogaraj@jrpsych.com.au; 5College of Engineering, Science and Environment, University of Newcastle, University Drive, Callaghan, NSW 2308, Australia; kim.colyvas@newcastle.edu.au

**Keywords:** mental health condition, chronic disease, health, sleep, smoking, nutrition, alcohol, physical activity

## Abstract

Introduction: People with a mental health condition experience disproportionate morbidity and mortality compared to the general population. This inequity has been largely attributed to a higher prevalence of chronic disease risk behaviours including smoking, poor nutrition, harmful alcohol consumption and inadequate physical activity (‘SNAP risks’). Suboptimal sleep is highly prevalent among people with a mental health condition and, as an identified risk behaviour for several chronic diseases, has been implicated as an additional contributor to this health inequity. Research involving people without a mental health condition suggests associations between poor sleep and each SNAP risk; however, interactions with mental health status have not been reported in an Australian population. This study explored associations between suboptimal sleep and all four SNAP risks, and assessed whether they vary by mental health status. Materials and Methods: A descriptive study (*n* = 1265) was undertaken using self-report data from a cross-sectional telephone survey of Australian adults. Based on national guidelines and recommendations that indicate when someone might be at risk of adverse health effects, SNAP risks and sleep variables were reduced to two levels: ‘at risk’ or ‘not at risk’; and ‘appropriate’ or ‘suboptimal’, respectively. Chi square tests and multivariable logistic regression models explored associations between suboptimal sleep, SNAP risks and mental health status. Results: Fifteen per cent (*n* = 184) of participants identified as having a mental health condition in the past 12 months. Being at risk of adverse health effects due to smoking had the strongest association with several measures of suboptimal sleep (*ps* < 0.05). Two-way interactions revealed that being at risk of adverse health effects due to alcohol use and physical inactivity resulted in a significantly greater likelihood of suboptimal sleep duration (OR 3.06, 95% CI 1.41 to 6.64; OR 3.06, 95% CI 1.41 to 6.69) and nap duration (OR 7.96, 95% CI 1.90 to 33.22), respectively, for people with a mental health condition compared to those without. Conclusions: The findings suggest associations between suboptimal sleep and smoking, risky alcohol consumption and physical inactivity, with the latter two perhaps being stronger among people with a mental health condition compared to those without such a condition. Poor sleep should be considered in interventions to address smoking, alcohol and physical activity; and vice versa. This study lends further support for the value of multirisk lifestyle interventions to promote physical and mental health for people with mental health conditions.

## 1. Introduction

Chronic diseases (such as cardiovascular disease, cancer and diabetes) account for more than 80% of the total disease burden in high-income countries including Australia, the United Kingdom (UK) and the Unites States (USA) [1]. A third of this burden could be prevented by reducing the prevalence of four modifiable chronic disease risk behaviours: smoking, poor nutrition, harmful alcohol consumption and inadequate physical activity (i.e., ‘SNAP risks’) [2]. SNAP risks often co-occur, and the presence of one risk behaviour, in some instances, has been shown to increase the likelihood that an individual will have one of the other risk behaviours [3].

Recent evidence has demonstrated that poor sleep (defined as sleep of insufficient or excessive duration and/or poor quality) also significantly increases the risk of developing various prevalent chronic diseases [4,5]. As a result, the need to consider poor sleep as an additional modifiable risk behaviour for chronic disease has recently been recognised [6]. This recognition was closely followed by international acknowledgement of sleep as the third ‘pillar’ of health, alongside a healthy diet and regular physical activity [7] and Australia’s parliamentary inquiry into the sleep health of the nation [8].

In addition to being a risk factor for chronic disease in and of itself, it is likely that poor sleep also acts synergistically to increase the disease burden related to other risk behaviours, with evidence suggesting associations between poor sleep and SNAP risks [9,10,11,12,13,14,15,16,17]. For example, a longitudinal study of youth (recruited in 10th grade; N = 1394) surveyed annually between 2009 and 2012 in the US found reciprocal and prospective relationships between poor sleep and smoking [14], where poor sleepers were more likely to start smoking and smokers were more likely to develop poor sleep. Furthermore, a prospective longitudinal study, undertaken between 2000 and 2007, involving 8960 Finnish adults found poor sleep at baseline was associated with 1.3 times greater odds of heavy drinking and physical inactivity at the 5 to 7 year follow-up [18]. Additionally, heavy drinking and binge drinking at baseline were associated with 1.5 and 1.3 greater odds, respectively, of poor sleep at follow-up [18]. Lastly, a cross sectional study of middle-aged Japanese females (N = 3129) found that, after controlling for known confounders, poor sleep quality was associated with low vegetable and fish consumption and high intake of confectionary and energy drinks (*p* < 0.05) [16].

People with a mental health disorder in Australia have a reduced life expectancy compared to the general population: 16 years less for men and 12 years less for women [19], with 78% of this excess mortality attributed to preventable chronic diseases including cancers, cardiovascular disease and respiratory disease [20]. The prevalence of SNAP risks is also comparatively high among this group compared to people without a mental health condition [21,22]. For example, recent research among 558 community mental health clients in New South Wales (NSW) found: 51% were smokers; 78% and 60% consumed less than the daily recommended amount of vegetables and fruit, respectively; 40% were at risk of acute alcohol related harm; and the same proportion did not meet the minimum recommended amount of daily physical activity [22]; with the majority (78%) meeting criteria for two or more risks [22]. With the exception of fruit and vegetable consumption, these reported prevalence was consistently almost double those found for clients attending general (non-mental health) community health services in the same area [23].

Poor sleep is also more prevalent among people with a mental health condition: reported to be up to 50–80% of individuals depending on psychiatric diagnosis and setting [20], compared to approximately 40% of individuals in the general population [24]. Recent systematic reviews [20] and meta reviews [25] published in high impact journals including *Lancet Psychiatry* and *World Psychiatry* have confirmed that poor sleep substantively contributes to both the excess physical health burden experienced by people with a mental health condition [20] and the development and maintenance of mental health disorders [25].

However, limited research has explored associations between sleep and SNAP risks among people with a mental health condition, with existing studies focussed only on associations between physical activity and sleep [26,27]. Most recently, a systematic review and meta-analysis of exercise interventions for improving sleep quality among people with various severe mental health conditions found that overall, increased exercise had a large statistically significant positive effect on sleep quality [27]. Another study, a two-month non-randomised trial involving participants with Bipolar Disorder (*n* = 32) and participants with no mental health condition (*n* = 36) from the UK, found similar bidirectional links between sleep and physical activity for both groups: for every standard deviation increase in poor sleep, there was approximately a 3% decrease in subsequent daily physical activity; whilst increased minutes of physical activity per day were associated with improved sleep [26].

Differences in associations between SNAP risks and sleep for people with and without a mental health condition might be expected, given the substantially higher prevalence of SNAP risks among people with a mental health condition [22] and the intrinsic link between mental health and sleep [20,28]. Understanding the relationships between sleep and all four SNAP risks, and potential variations according to mental health status, may help to better inform the development of public health interventions that could target multiple chronic disease risk behaviours (i.e., ‘multirisk lifestyle interventions’). Multirisk lifestyle interventions have been highlighted as a priority for future intervention research for the general population [29] and to redress the physical health inequities experienced by people with a mental health condition [20].

To address these gaps in the literature, this study recruited community dwelling persons in Australia with or without a mental health condition in the past 12 months in order to explore (1) associations between sleep and all four SNAP risks, and (2) assess whether associations varied by mental health status.

## 2. Material and Methods

### 2.1. Design and Setting

A subset of data was analysed from an Australia-wide cross-sectional telephone survey, the National Social Survey (NSS). The NSS is undertaken annually by the Population Research Laboratory at Central Queensland University in Australia and is comprised of items assessing participant demographic information, health behaviours, chronic disease status, and quality of life.

### 2.2. Data Collection

Trained interviewers, supported by a Computer-Assisted Telephone Interviewing (CATI) system at Central Queensland University, administered the NSS. The interviews were undertaken between July and August in 2017 at various times throughout the day, seven days per week. A minimum of six call attempts were made to every randomly selected number. The average interview length was 38 min.

### 2.3. Sample and Recruitment

Dual frame (mobile and landline; 1:1 ratio) random digit dialling was employed and random number selection approaches were used to ensure all community dwelling adults had an equal chance of being contacted. Eligibility criteria for the survey were being: an Australian resident, 18 years of age or older, and contactable by landline or mobile telephone. For mobile telephone numbers, the person who received the call was the respondent. For landline numbers, a respondent within each household was selected using the following process: (1) each landline number was randomly pre-selected to target either a male or female, (2) the dwelling was confirmed to be the person’s usual place of residence, (3) if there was more than one male/female in the household then the person of the target sex with the most recent birthday was selected, and (4) if there was no-one of the targeted sex residing in the dwelling it was designated ‘not qualified’.

Verbal informed consent was obtained prior to progressing with the interview.

### 2.4. Measures

The subset of survey items assessed as part of the current study included: participant demographic information (including mental health status); sleep parameters; and SNAP risk behaviours.

#### 2.4.1. Participant Demographic Information and Mental Health Status

Demographic information collected included: sex (male, female), age (years), marital status (single, widowed, divorced, separated not divorced, married, de facto), employment status (full-time, part-time/casual, unemployed, retired/pension, student, home duties), highest level of education (preschool, primary school, secondary school, technical or further educational institution, university or other higher education, none), Aboriginal and/or Torres Strait Islander origin (yes, no), geographic location (city, town, rural area) [30] and state or territory of residence (selected one of the eight states/territories of Australia).

The following item assessed mental health status: ‘Have you received a diagnosis or treatment for any of the following mental health conditions in the past 12 months: depression, anxiety, schizophrenia, other form of psychosis, bipolar disorder, personality disorder, substance use disorder, other mental health condition, none?’ [30,31]. Multiple responses were permitted.

#### 2.4.2. Sleep Parameters

Separate items assessed ‘sleep duration’ on weekdays and/or workdays (hours) and weekends and/or nonwork days (hours). Measures of ‘sleep quality’ included: sleep onset latency (minutes); number and duration (minutes) of awakenings throughout night; sleep efficiency (%); and number (per day) and duration (minutes) of naps [31]. These items have been utilised previously [31] and asked participants to consider their sleep behaviour on either an average day or week.

#### 2.4.3. SNAP Risks

To assess smoking status, participants were asked ‘Are you presently a smoker?’ (yes, no, do not know). If participants responded ‘yes’, they were asked how many cigarettes per day they smoke (<1, 1–5, 6–10, 11–15, 16–20, 21–30, 31–40, >40). In terms of nutrition, two items assessed the number of serves of fruit (item 1) and vegetables (item 2) consumed on a usual day [32]. To facilitate accurate responding, a description of a ‘serve’ of fruit and vegetables was verbally provided by the interviewer. Alcohol consumption was assessed using The AUDIT Alcohol Consumption Questions (AUDIT-C; 3 items) [33]. Physical activity was assessed via items 1 to 8 of The Active Australia Survey [34], which assesses, in the past week, the number of times respondents engaged in 10+ minutes of walking, vigorous gardening or heavy yard work, and vigorous and moderate physical activity. It also assesses total time engaged in each activity across the week.

### 2.5. Variable Transformation

In accordance with the National Sleep Foundation (NSF) sleep duration and quality criteria [35,36], data for sleep parameters were categorised as ‘inappropriate’/‘not recommended’ (hereafter referred to as ‘suboptimal’), ‘may be appropriate’/‘uncertain’ (hereafter referred to as ‘may be appropriate’) or ‘appropriate’/‘recommended’ (hereafter referred to as ‘appropriate’) (see Appendix A) [31]. Sleep duration and quality criteria are different for people aged 18 to 25, 26 to 64 and 65 or older (see Appendix A). Sleep measures were dichotomised to the categories ‘suboptimal’ and ‘may be appropriate/appropriate’ for inclusion in the association analyses.

Based on national guidelines and recommendations which indicate when someone might be at risk of adverse health effects, SNAP risks were reduced to two levels, categorised as ‘at risk’ or ‘not at risk’:

Smoking: participants that reported smoking any number of cigarettes per day or week were considered to be ‘at risk’ for tobacco related harm [37]. Nonsmokers were coded as ‘not at risk’. 

Nutrition: daily fruit and vegetable consumption was categorised according to Australian Dietary Guidelines [38], with participants consuming the guideline recommended minimum number of serves of fruit (2 serves) and vegetables (5 serves) per day categorised as ‘not at risk’, and those consuming less than recommended serves categorised as ‘at risk’. 

Alcohol: in accordance with AUDIT-C scoring guidelines, scores of ≥4 and ≥3 for men and women (indicating more than two standard drinks per day, or more than four standard drinks on any one occasion), respectively, were categorised as being ‘at risk’ for heavy drinking and/or active alcohol abuse or dependence [33]. Those with lower scores were ‘not at risk’. 

Physical activity: daily physical activity levels were categorised according to the Australian Physical Activity and Sedentary Behaviour Guidelines [39], with participants meeting guideline recommendations (150 min of moderate or 75 min of vigorous physical activity per week) considered ‘not at risk’, and those below recommendations ‘at risk’.

The following demographic and clinical variables were reduced to two or three levels for the purpose of association analyses: mental health status (mental health condition, no mental health condition), marital status (partnered, not partnered), age (18–25, 26–64, 65+) [35,36], identified to be Aboriginal and/or Torres Strait Islander (yes, no/no response), education (up to secondary school, technical/tertiary education), employment (paid employment, no paid employment), geographic distribution (city, town, rural area).

### 2.6. Analyses

Analyses were conducted using IBM SPSS Statistics 24 [40] (IBM, Armonk, NY, USA). Descriptive statistics were used to summarise participant demographic information and the prevalence of suboptimal sleep and SNAP risks, according to mental health status. Chi square analyses and independent samples t-tests assessed for differences in demographics, sleep parameters and SNAP risks according to mental health status.

Aim 1: Univariate associations between sleep parameters and SNAP risks were explored using chi squared test or Fischer’s exact test (where applicable). Multivariable logistic regression analyses, using the enter method, then explored relationships between sleep parameters and all four SNAP risks and to provide an adjusted analysis. Seven logistic regression models were developed for the following outcomes: sleep duration on weekdays and weekends, sleep onset latency, awakenings, wake after sleep onset, sleep efficiency, nap duration. Models were not developed for ‘days per week take a nap’ and ‘number of naps per day’ due to absence of ‘suboptimal criteria’ and <1% of participants meeting suboptimal criteria, respectively. To control for the known effects of age and sex [24,31], both variables were entered and retained in all models. Odds ratios and 95% confidence intervals were reported.

Aim 2: To explore if associations varied according to mental health status, building on the original logistic regression models developed to address Aim 1, all two-way interactions between mental health status and each SNAP risk were simultaneously entered into the model for each of the seven sleep outcomes. The main effect for mental health status was also entered and retained in all models examining two-way interactions.

The critical *p* value was set at *p* ≤ 0.05 for all analyses.

## 3. Results

### 3.1. Sample

Of the 5450 people contacted by the interviewers, 1265 people consented to take part in the survey (response rate of 23%). Participant demographic and mental health status data are summarised in Table 1. One hundred and eighty four (15%) respondents identified to have been diagnosed with or treated for a mental health condition in the past 12 months, with anxiety (*n* = 108) and depression (*n* = 128) the most common conditions. There were significant differences between people with and without a mental health condition in terms of: sex (60% vs. 51% of people with a mental health condition were female; *χ*^2^(1) = 5.0, *p* = 0.03), age (73% vs. 64% were less than 65 years old *χ*^2^(2) = 6.3, *p* = 0.04), and employment status (60% vs. 43% were not employed full time; *χ*^2^(1) = 18.2, *p* < 0.001). There were no other significant differences between groups across other demographic characteristics.

### 3.2. Prevalence of Suboptimal Sleep and SNAP Risks

For both people with and without a mental health condition, the highest prevalence of suboptimal sleep was seen on measures of sleep duration and sleep onset latency (Table 2). Across most parameters (all except ‘number of naps per day’), the prevalence of suboptimal sleep was at least twice as high among people with a mental health condition compared to those without (Table 2).

People with a mental health condition were more likely than those without such a condition to be at of risk of adverse health effects due to smoking (22% vs. 11%) and physical inactivity (51% vs. 40%), but less likely to be at risk of alcohol-related harm (38% vs. 47%) (Table 3). The large majority of all participants were at risk of adverse health effects due to poor nutrition, with no difference between those with and without a mental health condition (90% vs. 92%, respectively; Table 3).

### 3.3. Associations between Suboptimal Sleep and SNAP Risks

Table 4 summarises univariate associations between SNAP risks and suboptimal sleep. Across five of seven sleep parameters (except Wake After Sleep Onset and Sleep Efficiency), being at risk of adverse health effects due to smoking was associated with a significantly higher odds of experiencing suboptimal sleep. Multivariable modelling reflected a similar pattern, with risk for smoking associated with suboptimal sleep across the same five parameters while also indicating risk of adverse health effects due to physical inactivity was associated with a higher likelihood of suboptimal wake after sleep onset (OR 1.6, 95% CI 1.01 to 2.51) (Table 5).

### 3.4. Variations in Associations between Suboptimal Sleep and SNAP Risks According to Mental Health Status

Differences in the strength of associations between a SNAP risk and suboptimal sleep for people with a mental health condition compared to those without were found on three sleep parameters: sleep duration on weekdays, sleep duration on weekends and nap duration (Appendix A). For sleep duration on weekdays and on weekends, a significant interaction between being at risk of adverse health effects due alcohol and mental health status was found. Participants at risk due to alcohol consumption and who had a mental health condition had 3.06 times the odds.

(95% CI for weekdays: 1.41 to 6.64; 95% CI for weekends: 1.41 to 6.69) of meeting the criteria for suboptimal sleep duration compared to people without a mental health condition. Among those not at risk of adverse health effects due to alcohol consumption, there were no difference on weekdays or weekends among people with and without a mental health condition in terms of meeting suboptimal criteria for sleep duration (weekdays: OR 1.60, 95% CI 0.73 to 3.49; weekends: OR 1.60, 95% CI 0.73 to 3.52).

Furthermore, for nap duration, there was a significant interaction between risk due to physical inactivity and mental health status. For participants at risk of adverse health effects due to physical inactivity, people with a mental health condition had nearly eight times the odds (OR 7.96, 95% CI 1.90 to 33.22) of meeting the criteria for suboptimal nap duration compared to people without such a condition. For those not at risk due to physical inactivity, there were no differences between people with and without a mental health condition in terms of meeting suboptimal criteria for nap duration (OR 0.89, 95% CI 0.17 to 4.57).

## 4. Discussion

This is the first Australian study to explore associations between sleep and all four SNAP risks, and to assess if they vary among people with and without a mental health condition. Reflecting previous research, the prevalence of smoking, physical inactivity and poor sleep were higher for people with a mental health condition compared to those without [21,22,23]. For both groups, across the majority of the sleep parameters, being a smoker was the strongest predictor of suboptimal sleep. On measures of sleep and nap duration, risky alcohol consumption and physical inactivity, respectively, led to significantly greater odds of suboptimal sleep for people with a mental health condition compared to those without such a condition. Our findings suggest associations between suboptimal sleep and smoking, risky alcohol consumption and physical inactivity, with the strength of the latter two associations perhaps stronger among people with a mental health condition compared to those without such a condition. The results lend further support for the value of multirisk lifestyle interventions to promote physical and mental health among people with a mental health condition.

The finding that smoking increased the likelihood of suboptimal sleep across the majority of parameters is consistent with previous research in the US [14] and Germany [17]. Using polysomnography, Jaehne et al. [17] found adult smokers, compared to nonsmokers, had shorter total sleep time, longer sleep onset latency, and more sleep apnoea and leg movements; with a positive dose–response relationship observed between nicotine dependence/plasma nicotine concentration and extent of sleep issues. While further research is required, the authors suggested potential mechanisms underlying the association between smoking and suboptimal sleep may include the arousal-like effect of nicotine due to cholinergic stimulation, effects of nicotine withdrawal, and/or a reduction in hypoxic sensitivity and a delay in hypoxia-induced arousals [17]. There are complex relationships between mental health, smoking [41,42] and sleep [43,44], so research investigating the specific mechanisms underpinning suboptimal sleep among smokers with a mental health condition, including the presence of dose–response relationship, is needed. Furthermore, given the markedly high prevalence of smoking among people with a mental health condition, addressing smoking is likely of importance in interventions to address sleep specifically and also in multirisk lifestyle interventions for this group. Sleep optimisation may also be a novel addition to existing smoking cessation interventions.

Our findings further suggest the impact of risky alcohol consumption and physical inactivity on sleep may be stronger for people with a mental health condition compared to those without a mental health condition. With regard to risky alcohol consumption, among those that were ‘at risk’ of adverse health effects, the likelihood of suboptimal sleep duration was more than three times higher than among those without such a condition, suggesting risky alcohol use has a greater impact on sleep duration for people with a mental health condition. Review evidence suggests variable and dose-related effects of alcohol on sleep duration and continuity, with lower doses increasing sleep time and higher doses linked to increased awakenings, likely due to short-term withdrawal, which increases sympathetic activity, especially during the second half of the night [45]. People with a mental health condition typically experience more difficulties with sleep onset and maintenance compared to those without a mental health condition [46], so it could be the effects of risky alcohol consumption exacerbate these existing difficulties, which in turn lead to suboptimal sleep duration. A paucity of research has explored mechanisms underpinning the relationship between suboptimal sleep duration (or poor sleep more generally) and risky alcohol use among people with a mental health condition [45,47]. A US study reported similar REM sleep changes during withdrawal among people with risky alcohol use with and without a secondary depressive disorder, which may suggest such disorders do not moderate the impact of alcohol on REM sleep [47]. The findings highlight the need to consider and address risky alcohol use in sleep interventions, and vice versa, particularly for people with a mental health condition.

In terms of physical inactivity, people with a mental health condition who were at risk of adverse health effects due to physical inactivity were more likely to take longer than recommended naps compared to those at risk without a mental health condition. However, there were no differences between people with and without a mental health condition for those who reported being sufficiently active. Common reasons for napping include recovering lost sleep (‘recovery nap’) and increasing energy levels/improving mood (‘appetitive nap’) [48]. It is well known that low energy and mood and feelings of fatigue are symptoms of various mental health conditions [49] for which physical activity is an effective treatment [25,50]. Our results may therefore suggest that when people with a mental health condition are not sufficiently active, they may rely more heavily on naps to manage symptoms of fatigue and mood difficulties and to boost energy levels. However, when they are active, they do not need to engage in naps of suboptimal duration. This hypothesis requires further exploration.

Contrary to previous research [51,52], we found no association between measures of suboptimal sleep and nutrition. The large majority (>90%) of participants had insufficient fruit and vegetable consumption, according to national guidelines [38]. While categorising risk of adverse health effects due to poor nutrition according to national guidelines is recommended, valid and reliable, it lacks specificity when exploring relationships with suboptimal sleep. For example, research suggests diet-related factors other than quantity of types of food, such as regularity and timing of meals, significantly impact on sleep quality [53]. Furthermore, in addition to fruit and vegetables, the consumption of other food groups impact sleep, with recent research suggesting adherence to a Mediterranean diet is associated with better quality sleep [51]. Future research exploring associations between sleep and diet among people with and without a mental health condition should consider using a more comprehensive measure of dietary intake and also explore the impact of meal regularity and timing.

The findings of the current study need be considered in the context of a number of methodological characteristics. Firstly, the study relied on self-report data. With regard to sleep measures, while there has been suggestion of the reduced accuracy of self-report data for sleep parameters [54], studies indicate that such data are equally predictive of sleep-related morbidity and mortality when compared to more objective measures [55,56,57]. The collection of objective sleep data would strengthen the conclusions able to be drawn from future surveys. The item used to determine overall mental health status has established validity and has been used in previous population-level health surveys assessing chronic disease risks [30,58]. However, its precision in differentiating between specific mental health disorders (e.g., depression vs. bipolar disorder) has not been established. The questions used to assess SNAP risks have been validated against objective measures [32,34] and align directly with national guidelines/recommendations [38,39]. Nevertheless, they preclude assessment of the impact of other factors such as timing of smoking and alcohol use per day or consumption of foods other than fruit and vegetables, which may be important when exploring associations with sleep.

Next, the use of a telephone survey likely prevented clinical mental health populations (i.e., those in inpatient units) from participating and resulted in a low response rate. In terms of the former, future research exploring associations between SNAP risks and sleep among people with a mental health condition should consider methodology that ensures such groups can participate. With regard to the latter, while the response rate is low, it is comparable to other national telephone surveys [59] and to that achieved previously for the NSS [60]. In terms of sample representativeness, older adults (65+ years) and those living outside of urban areas were overrepresented, compared to Australian Bureau of Statistics data. Nevertheless, the impact on our findings was likely negligible, with nonsignificant differences found between outcomes using raw data and weighted data (accounting for overrepresentation).

Lastly, the findings need to be interpreted in the context of the cross-sectional study design. To assess bidirectionality between sleep and SNAP risks, and variations according to mental health status, future research employing a longitudinal design is needed. Such research may also consider three-way interactions between sleep, mental health and each SNAP risk.

## 5. Conclusions

Our findings suggest associations between suboptimal sleep and smoking, risky alcohol consumption and physical inactivity, with the strength of the latter two associations perhaps stronger among people with a mental health condition compared to those without such a condition. Poor sleep should be considered in interventions to address smoking, risky alcohol use and physical inactivity; and vice versa. Further research on the relationship between nutrition and sleep is needed. Overall, this study lends further support for the value of multirisk lifestyle interventions to promote physical and mental health among people with a mental health condition.

## Figures and Tables

**Table 1 ijerph-18-05946-t001:** Demographic Information and Health Status for Participants with and without a Mental Health Condition.

Demographic Information	People with a MHC (*n* = 184)	People without a MHC (*n* = 1081)	Total (*n* = 1265)
	*n* (%)	*n* (%)	*n* (%)
**Marital status**			
Single (never married)	44 (23.9)	189 (17.5)	233 (18.4)
Widowed	18 (9.8)	88 (8.1)	106 (8.4)
Divorced	15 (8.2)	53 (4.9)	68 (5.4)
Separated not divorced	13 (7.1)	21 (1.9)	34 (2.7)
Married	74 (40.2)	639 (59.1)	713 (56.4)
De facto	18 (9.8)	86 (8.0)	104 (8.2)
No response	2 (1.1)	5 (0.5)	7 (0.6)
**Sex**			
Male	73 (39.7)	525 (48.6)	598 (47.3)
**Age**			
18–25	20 (10.9)	94 (8.7)	114 (9.0)
26–64	114 (62.0)	586 (54.2)	700 (55.3)
65+	50 (27.2)	393 (36.4)	443 (35.0)
No response	0 (0.0)	8 (0.7)	8 (0.6)
**Identify to be Aboriginal and/or Torres Strait Islander**			
Yes	5 (2.7)	22 (2.0)	27 (2.1)
No	178 (96.7)	1054 (97.5)	1232 (97.4)
Unsure/no response	1 (0.5)	5 (0.5)	6 (0.05)
**Highest level of education**			
Preschool	0 (0.0)	3 (0)	3 (0)
Infants/primary school	3 (1.6)	17 (2)	20 (2)
Secondary/high school	62 (33.7)	332 (31)	394 (31)
Technical or further educational institution (e.g., TAFE colleges)	32 (17.4)	211 (20)	243 (19)
University or other higher educational institution	86 (46.7)	512 (47)	598 (47)
No schooling/no response	1 (0.05)	6 (1)	7 (1)
**Employment status** ^&^			
Employed full-time	42 (22.8)	393 (36.4)	435 (34.4)
Employed part-time/casual	32 (17.4)	223 (20.6)	255 (20.2)
Unemployed	20 (10.9)	35 (3.2)	55 (4.3)
Retired/pension	73 (39.7)	383 (35.4)	456 (36.0)
Student	6 (3.3)	23 (2.1)	29 (2.3)
Home duties	8 (4.3)	17 (1.6)	25 (2.0)
No response	3 (1.6)	7 (0.6)	10 (0.8)
**State or territory residing**			
Australian Capital Territory (ACT)	0 (0.0)	19 (1.8)	19 (1.5)
New South Wales (NSW)	51 (27.7)	323 (29.9)	374 (30.0)
Northern Territory (NT)	1 (0.05)	11 (1.0)	12 (0.9)
Queensland (QLD)	45 (24.5)	236 (21.8)	282 (22.3)
South Australia (SA)	20 (10.9)	86 (8.0)	106 (8.4)
Tasmania (TAS)	4 (2.2)	30 (2.8)	34 (2.7)
Victoria (VIC)	39 (21.2)	268 (24.8)	307 (24.3)
Western Australia (WA)	23 (12.5)	104 (9.6)	127 (10.0)
No response	0 (0.0)	4 (0.4)	4 (0.3)
**Geographic distribution**			
City	100 (54.3)	562 (51.8)	662 (52.3)
Town	34 (18.5)	224 (20.7)	278 (22.0)
Rural area	49 (26.6)	271 (25.1)	320 (25.3)
Unsure/ no response	1 (0.5)	4 (0.4)	5 (0.4)
**Type of mental health condition (12-month)** ^#^			
Depression	128 (70.0)	-	128 (10.1)
Anxiety disorder	108 (58.7)	-	108 (8.5)
Schizophrenia	6 (3.3)	-	6 (0.5)
Bipolar disorder	9 (4.9)	-	9 (0.7)
Personality disorder	7 (3.8)	-	7 (0.6)
Other form of psychosis	3 (1.6)	-	3 (0.2)
Substance use disorder	2 (1.1)	-	2 (0.2)
Other mental health condition	20 (10.9)	-	18 (1.4)
None of the above	N/A	-	1058 (83.6)
No response	0 (0.0)	-	23 (1.8)
**Number of mental health conditions**			
0	0 (0.0)	-	1081 (85.5)
1	105 (57.1)	-	105 (8.3)
2	64 (34.8)	-	64 (5.1)
3	12 (6.5)	-	12 (0.9)
4	3 (1.6)	-	3 (0.2)

^&^ Primary employment status; ^#^ Multiple responses permitted; N/A: Not applicable; MHC: Mental health condition.

**Table 2 ijerph-18-05946-t002:** Prevalence of Suboptimal Sleep among People with and without a Mental Health Condition.

		Descriptive Statistics				Association Analyses			
		People with a MHC	People without a MHC	Total		People with a MHC	People without a MHC	Total	
		*n* (%)	*n* (%)	*n* (%)		*n* (%)	*n* (%)	*n* (%)	
**Sleep duration** (weekdays) ^1^									*χ*^2^(1) = 31.1 ***
Appropriate		62 (33.9)	418 (39.4)	480 (38.6)	Appropriate/May be appropriate	112 (62.1)	848 (79.9)	960 (77.2)	
May be appropriate		50 (27.3)	430 (40.5)	480 (38.6)					
Suboptimal					Suboptimal	71 (38.8)	213 (20.1)	284 (22.8)	
	*Inadequate*	*55* (*77.5*)	*175* (*82.2*)	*230* (*81.0*)					
	*Excessive*	*16* (*22.5*)	*38* (*17.8*)	*54* (*19.0*)					
**Sleep duration** (weekends) ^1^									*χ*^2^(1) = 37.0 ***
Appropriate		60 (33.0)	515 (48.5)	575 (46.2)	Appropriate/May be appropriate	116 (63.7)	883 (83.1)	999 (80.3)	
May be appropriate		56 (30.8)	368 (34.7)	424 (34.1)					
Suboptimal					Suboptimal	66 (36.3)	179 (16.9)	245 (19.7)	
	*Inadequate*	*48* (*72.7*)	*127* (*70.9*)	*175* (*71.4*)					
	*Excessive*	*18* (*27.3*)	*52* (*29.1*)	*70* (*28.6*)					
**Sleep onset latency** ^2^									*χ*^2^(1) = 32.4 ***
Appropriate		118 (65.9)	874 (81.8)	992 (79.6)	Appropriate/May be appropriate	129 (72.1)	941 (88.1)	1070 (85.8)	
May be appropriate		11 (6.1)	67 (6.3)	78 (6.3)					
Suboptimal		50 (27.9)	127 (11.9)	177 (14.2)	Suboptimal	50 (27.9)	127 (11.9)	177 (14.2)	
**Awakenings (>5 min)** ^3^									*χ*^2^(1) = 19.0 ***
Appropriate		90 (49.5)	719 (67.1)	809 (64.5)	Appropriate/May be appropriate	149 (81.9)	987 (92.1)	1136 (90.6)	
May be appropriate		59 (32.4)	268 (25.0)	327 (26.1)					
Suboptimal		33 (18.1)	85 (7.9)	118 (9.4)	Suboptimal	33 (18.1)	85 (7.9)	118 (9.4)	
**Wake after sleep onset (duration)** ^1^									*χ*^2^(1) = 13.1 ***
Appropriate		130 (71.8)	894 (84.1)	1024 (82.3)	Appropriate/May be appropriate	159 (87.8)	999 (94.0)	1158 (93.1)	
May be appropriate		29 (16.0)	105 (9.9)	134 (10.8)					
Suboptimal		22 (12.2)	64 (6.0)	86 (6.9)	Suboptimal	22 (12.2)	64 (6.0)	86 (6.9)	
**Sleep efficiency** ^&,4^									*χ*^2^(1) = 9.0 **
Appropriate		119 (71.3)	845 (83.5)	964 (81.8)	Appropriate/May be appropriate	146 (87.4)	959 (95.8)	1105 (93.7)	
May be appropriate		27 (16.2)	114 (11.3)	141 (12.0)					
Suboptimal		21 (12.6)	53 (5.2)	74 (6.3)	Suboptimal	21 (12.6)	53 (5.2)	74 (6.3)	
**Nap frequency (days/week)** ^5^									
Appropriate		13 (7.1)	55 (5.1)	68 (5.4)					
May be appropriate		171 (92.9)	1021 (94.9)	1192 (94.6)					
Suboptimal		N/A	N/A	N/A					
**Number of naps/day** ^6^									*χ*^2^(1) = 0.1
Appropriate		13 (7.1)	55 (5.1)	68 (5.4)	Appropriate/May be appropriate	183 (99.5)	1066 (99.3)	1249 (99.3)	
May be appropriate		170 (92.4)	1011 (94.1)	1181 (93.9)					
Suboptimal		1 (0.5)	8 (0.7)	9 (0.7)	Suboptimal	1 (0.5)	8 (0.7)	9 (0.7)	
**Nap duration** ^7^									*χ*^2^(1) = 17.2 ***
Appropriate		N/A	N/A	N/A	Appropriate/May be appropriate	166 (90.2)	1035 (96.8)	1201 (95.8)	
May be appropriate		166 (90.2)	1035 (96.8)	1201 (95.8)					
Suboptimal		18 (9.8)	34 (3.2)	52 (4.2)	Suboptimal	18 (9.8)	34 (3.2)	52 (4.2)	

^1^ Missing: *n* = 21; ^2^ Missing: *n* = 18; ^3^ Missing: *n* = 11; ^4^ Missing: *n* = 86; ^5^ Missing: *n* = 5;^6^ Missing: *n* = 7; ^7^ Missing: *n* = 12; ^&^ Sleep efficiency on weekdays; N/A: not applicable; MHC: mental health condition. ** *p* < 0.01; *** *p* < 0.001.

**Table 3 ijerph-18-05946-t003:** Prevalence of SNAP Risks among People with and without a Mental Health Condition.

SNAP Risks	People with a MHC	People without a MHC	Total	
	*n* (%)	*n* (%)	*n* (%)	
**Smoking**				*χ*^2^(1) = 17.59 ***
At risk	41 (22.3)	123 (11.4)	162 (12.8)	
Not at risk	143 (77.7)	958 (88.6)	1101 (87.0)	
**Nutrition**				*χ*^2^(1) = 0.84
At risk	170 (92.4)	972 (89.9)	1142 (90.3)	
Not at risk	14 (7.6)	105 (9.7)	119 (9.4)	
**Alcohol**				*χ*^2^(1) = 5.08 *
At risk	70 (38.0)	508 (47.0)	578 (45.7)	
Not at risk	114 (62.0)	573 (53.0)	685 (54.2)	
**Physical activity**				*χ*^2^(1) = 7.50 **
At risk	93 (50.5)	434 (40.1)	527 (41.7)	
Not at risk	91 (49.5)	647 (59.9)	738 (58.3)	

MHC: Mental health condition; * *p* < 0.05; ** *p* < 0.01; *** *p* < 0.001.

**Table 4 ijerph-18-05946-t004:** Univariate Associations between SNAP risks and Suboptimal Sleep.

SNAP Risks	Sleep Duration (Weekdays) ^1^	Sleep Duration (Weekends) ^1^	Sleep Onset Latency ^1^	Awakenings ^1^	Wake After Sleep Onset ^1^	Sleep Efficiency ^1^	Nap Duration ^1^
	*n* (%)	*n* (%)	*n* (%)	*n* (%)	*n* (%)	*n* (%)	*n* (%)
**Smoking**							
At risk	50 (31.3) **	50 (31.1) ***	35 (22.2) **	24 (14.9) **	13 (8.2)	12 (8.5)	18 (11.3) ***
Not at risk	234 (21.6)	195 (18.0)	142 (13.1)	94 (8.6)	73 (6.7)	62 (6.0)	34 (3.1)
**Nutrition**							
At risk	255 (22.6)	219 (19.4)	164 (14.5)	105 (9.3)	81 (7.2)	69 (6.4)	48 (4.2)
Not at risk	28 (24.6)	25 (21.7)	11 (9.5)	13 (10.9)	5 (4.2)	5 (4.7)	4 (3.4)
**Alcohol**							
At risk	135 (23.7)	117 (20.4)	77 (13.5)	59 (10.3)	44 (7.7)	38 (7.0)	20 (3.5)
Not at risk	149 (22.1)	128 (19.1)	100 (14.8)	59 (8.7)	42 (6.2)	36 (5.7)	32 (4.7)
**Physical activity**							
At risk	119 (23.3)	110 (21.6)	71 (13.8)	53 (10.4)	43 (8.5)	37 (7.9)	25 (4.9)
Not at risk	165 (22.6)	135 (18.5)	106 (14.6)	65 (8.8)	43 (5.9)	37 (5.2)	27 (3.7)

** *p* <0.01; *** *p* <0.001. ^1^ Participants meeting criteria for ‘suboptimal’ sleep on this parameter.

**Table 5 ijerph-18-05946-t005:** Multivariable Associations between SNAP risks and Suboptimal Sleep.

SNAP Risks	OR	95% CI		*p*
		Lower	Upper	
**Sleep Duration (Weekdays)** ^1^				
Smoking ^2^	1.60	1.10	2.33	0.01
Nutrition ^2^	0.89	0.56	1.40	0.60
Alcohol ^2^	1.05	0.80	1.37	0.75
Physical Activity ^2^	1.04	0.79	1.37	0.80
**Sleep Duration (Weekends****)** ^1^				
Smoking ^2^	1.90	1.30	2.77	<0.001
Nutrition ^2^	0.82	0.51	1.32	0.42
Alcohol ^2^	1.02	0.77	1.36	0.88
Physical Activity ^2^	1.25	0.94	1.68	0.13
**Sleep Onset Latency** ^1^				
Smoking ^2^	1.69	1.09	2.63	0.02
Nutrition ^2^	1.68	0.87	3.25	0.12
Alcohol ^2^	0.94	0.67	1.33	0.73
Physical Activity ^2^	0.80	0.57	1.11	0.18
**Awakenings** ^1^				
Smoking ^2^	1.82	1.10	3.01	0.02
Nutrition ^2^	0.89	0.47	1.69	0.73
Alcohol ^2^	1.15	0.77	1.70	0.50
Physical Activity ^2^	1.13	0.77	1.67	0.53
**Wake After Sleep Onset** ^1^				
Smoking ^2^	1.03	0.54	1.94	0.94
Nutrition ^2^	1.67	0.66	4.26	0.28
Alcohol ^2^	1.28	0.82	2.01	0.28
Physical Activity ^2^	1.60	1.01	2.51	0.04
**Sleep Efficiency** ^1^				
Smoking ^2^	1.47	0.75	2.85	0.26
Nutrition ^2^	1.47	0.57	3.78	0.42
Alcohol ^2^	1.27	0.79	2.05	0.33
Physical Activity^2^	1.41	0.87	2.30	0.16
**Nap Duration** ^1^				
Smoking ^2^	4.01	2.15	7.50	<.001
Nutrition ^2^	1.09	0.37	3.15	0.88
Alcohol ^2^	0.57	0.31	1.04	0.07
Physical Activity ^2^	1.31	0.73	2.34	0.36

^1^ Reference category: Appropriate/may be appropriate. ^2^ Reference category: Not at risk.

## Data Availability

The materials and datasets generated and analysed during the current study are not publicly available. However, they are available from the corresponding author on reasonable request.

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
