# Peer review of "Associations between Suboptimal Sleep and Smoking, Poor Nutrition, Harmful Alcohol Consumption and Inadequate Physical Activity (‘SNAP Risks’): A Comparison of People with and without a Mental Health Condition in an Australian Community Survey"

_ijerph, 2021, doi:10.3390/ijerph18115946_

Round 1

Reviewer 1 Report

The paper presents the importance that people with mental disorders or not may interfere with the associations between poor sleep and SNAP risks. However, due to a low positive rate or small expectation in several variables/stratum, the analyses are suggested using more appropriate methods to make more scientific soundness. The article is recommended to reanalyze for more convincible results before having a further review. Please find my comments on the article in the notation of the attached file.

Reviewer 2 Report

The authors present results of an Austrian telephone population survey about associations between sleep and risk behaviors in people with and without a mental health condition, which has high clinical relevance. Their observations once again stress the need to include assessment and therapy of sleep disorders in clinical routine, which is often still neglected. In general, the paper is well-written, however, a few changes are needed to improve the understanding of the results and conclusions.

Abstract:

The abstract is rather long with an extensive introduction; however, study results are reported in a very superficial way, the effect of smoking for example is only characterised by a p-value, which has no informative value. This study has a cross-sectional design, which precludes causal interpretation and statements about directions of effects. Consequently, the phrasing “predictor” might be misleading and should be replaced.

Methods:

Replace “gender” by “sex” since gender refers to social identity.

Introduce abbreviations at their first mention in the text (e.g., NSF).

SNAP risk behaviors are assessed in a rather coarse-grained way. Include more fine-grained analyses are include this limitation in the discussion section.

In the statistical analyses, why you only include two-way interactions? Additionally, I do not understand when and why and how you use stepwise regression models to eliminate variables. You could just use the enter method and state all results, then the reader can evaluate which results are relevant or not. Since no power calculation is mentioned, statistical insignificance does not necessarily mean clinical irrelevance.

Results:

The response rate is rather low, is the sample still representative of the Australian general population?

Since you observe significant interactions with mental health status, I would recommend to also show analyses stratified by mental health status (concerns Table 4 and Table 5). Similarly, I would suggest showing results of Table 3 also stratified by mental health status and suboptimal sleep status, with suboptimal sleep status for example defined by suboptimal sleep in at least one of the domains.

Discussion:

You state that “A paucity of research has explored mechanisms underpinning the relationship between suboptimal sleep duration (or poor sleep more generally) and risky alcohol use among people with a mental health condition”, please discuss these research results.

Minor:

It has to read “Table 4 summarizes

The term “at risk for physical activity” in the results text (p.11) sounds awkward, I would suggest replacing “physical activity” by “physical inactivity”

Reviewer 3 Report

This is a very well written article describing the relationships among various healthy behavior risks, sleep quantity and quality, and the presence/absence of mental illness.  I have only a few comments that I would like for the authors to address. Once these issues are addressed, the paper should be published without further review in my opinion.

  1. Throughout the manuscript, the authors say things like "being at risk for smoking was the strongest predictor of xyz, and being at risk for alcohol use and physical inactivity resulted in an increased likelihood of xyz."  When first reading the manuscript, a reader will wonder how someone can be "at risk of smoking" or be "at risk of alcohol use" since people will either claim to be a smoker (or alcohol user) or claim not to be a smoker (or alcohol user). In the Methods section, the authors clarify that what they really mean to say is that "subjects are at risk of adverse health effects from smoking or alcohol use or inactivity (as is stated nicely throughout the Discussion section). Please go through the manuscript and rephrase all of the instances of these risk statements so they are easier for readers to correctly interpret.
  2. In section 2.3, the authors state "The process for randomly selecting respondents for landline telephone numbers is reported elsewhere [30]." Unfortunately, this reference is not freely accessible, so many readers may not be able to get it (at least right away). Therefore, I recommend that the authors state the process for randomly selecting respondents for landline telephone numbers in the present manuscript.  
  3. In the Discussion section dealing with nutrition, the following statement appears: "While this approach to assessing risk for poor nutrition is recommended, valid and reliable, it lacks specificity when exploring relationships with suboptimal sleep." Please be more definitive about exactly which "approach" is being discussed here.
  4. At the end of the Discussion section, the authors note some limitations as appropriate.  However, within this section, the authors state "The item used to determine mental health status has established validity and has been used in previous population-level health surveys assessing chronic disease risks [31,58]. Finally, questions used to assess SNAP risks have been validated against objective measures [32,34] and align directly to national guidelines/recommendations [38,39]." These statement appear to be factual evidence that the particular methods focused upon here are NOT limitations.  Thus, they should be removed from the limitations section unless they are rephrased to better fit into that section.

Round 2

Reviewer 1 Report

I think the authors have made a more reliable result after reanalyzing the data. Nonetheless, there are still some errors that need to be revised.

  1. As having the same problem as in Table 2 to Table4 in the original edition, Table 1 is suggested to recreate again after a reanalysis.
  2. Line 290. I cannot find 3.6 of odds ratio in Supplementary Table 2. Moreover, the statement may be wrong in the last sentence in the paragraph. I think it may happen due to an error of number (i.e. OR 1.60, 95% CI 1.21 to 1.96). 0.73 to 3.52 of 95% CI will be correct.

Author Response

We hope that we have addressed your additional comments, for which we are grateful.

Our responses and modifications to the manuscript are itemised below against each of the comments made. Revisions are highlighted using track changes throughout the manuscript.

1. As having the same problem as in Table 2 to Table 4 in the original edition, Table 1 is suggested to recreate again after a reanalysis.

We reduced variables in Table 1 for the purpose of association analyses. We have now included details of such transformations in the ‘Variable transformation’ section so it is clear what variables were entered into Chi-square analyses. Given the size of Table 1 and less direct alignment to study aims, we have opted to describe results of association analyses in the ‘Sample’ section of the results rather than represent in the Table as well. To enable readers to compare the demographics to the Australian population, we have a preference not to report on the reduced variables in the Table (if had to choose one approach or the other due to table size). We have deleted indications of significance (asterisks) from Table 1 to avoid confusing readers.

We can confirm, where applicable, no differences between results from Chi Square analyses and Fischer’s Exact test.

2. Line 290. I cannot find 3.6 of odds ratio in Supplementary Table 2.

 3.6 has been changed to 3.06, thanks for noticing this error.

3. Moreover, the statement may be wrong in the last sentence in the paragraph. I think it may happen due to an error of number (i.e. OR 1.60, 95% CI 1.21 to 1.96). 0.73 to 3.52 of 95% CI will be correct.

Error corrected, thanks for picking up.

Reviewer 2 Report

The authors adequately addressed my comments except for one comment: in supplementary Table 2 I prefer to see separate analyses for people with and without a mental health condition instead of using people without a mental health condition as reference. I agree that 3-way interactions are difficult to interpret but I wonder why they are not relevant for the present analyses, for example smoking often goes along with alcohol consumption in stressful conditions.

Author Response

We hope that we have addressed your additional comments, for which we are grateful.

Our responses and modifications to the manuscript are itemised below against each of the comments made. Revisions are highlighted using track changes throughout the manuscript.

  1. In supplementary Table 2 I prefer to see separate analyses for people with and without a mental health condition instead of using people without a mental health condition as reference.

Our study aims to explore (1) associations between sleep and all four SNAP risks, and (2) assess whether associations varied by mental health status. If we reverse the two-way interactions so ‘no risk’ for each SNAP risk was the reference category rather than ‘no mental health condition’, we would not be able to assess if the strength of the association between SNAP and sleep is statistically different or similar for people with a mental health condition compared to those without. Instead, we would be comparing the strength of association between mental health and sleep, contingent on risk status for SNAP risks, with our findings reflecting the impact of risk status for SNAP within people with a mental health condition (or those without), rather than between these groups.  As a result, we have a preference to keep data presented as is. However, of course, are happy to be guided by the editor in relation to this.

  1. I agree that 3-way interactions are difficult to interpret but I wonder why they are not relevant for the present analyses, for example smoking often goes along with alcohol consumption in stressful conditions.

While this analysis falls outside the scope of the current study aims, we have added a sentence to the Discussion suggesting it could be a focus of to future research.